# *HFE* genotypes, haemochromatosis diagnosis and clinical outcomes at age 80 years: a prospective cohort study in the UK Biobank

Mitchell R Lucas,[1] Janice L Atkins [ID],[1] Luke C Pilling [ID],[1] Jeremy D Shearman,[2] David Melzer[1]

MRL and JLA contributed equally.

¹Epidemiology and Public Health Group, Department of Clinical and Biomedical Sciences, Faculty of Health and Life Sciences, University of Exeter, Exeter, UK
²Department of Gastroenterology, South Warwickshire University NHS Foundation Trust, Warwick, UK

**Correspondence to**
Professor David Melzer;
d.melzer@exeter.ac.uk

## ABSTRACT

**Objectives** *HFE* haemochromatosis genetic variants have an uncertain clinical penetrance, especially to older ages and in undiagnosed groups. We estimated p.C282Y and p.H63D variant cumulative incidence of multiple clinical outcomes in a large community cohort.

**Design** Prospective cohort study.

**Setting** 22 assessment centres across England, Scotland, and Wales in the UK Biobank (2006–2010).

**Participants** 451 270 participants genetically similar to the 1000 Genomes European reference population, with a mean of 13.3-year follow-up through hospital inpatient, cancer registries and death certificate data.

**Main outcome measures** Cox proportional HRs of incident clinical outcomes and mortality in those with *HFE* p.C282Y/p.H63D mutations compared with those with no variants, stratified by sex and adjusted for age, assessment centre and genetic stratification. Cumulative incidences were estimated from age 40 years to 80 years.

**Results** 12.1% of p.C282Y+/+ males had baseline (mean age 57 years) haemochromatosis diagnoses, with a cumulative incidence of 56.4% at age 80 years. 33.1% died vs 25.4% without *HFE* variants (HR 1.29, 95% CI: 1.12 to 1.48, p=4.7×10⁻⁴); 27.9% vs 17.1% had joint replacements, 20.3% vs 8.3% had liver disease, and there were excess delirium, dementia, and Parkinson's disease but not depression. Associations, including excess mortality, were similar in the group undiagnosed with haemochromatosis. 3.4% of women with p.C282Y+/+ had baseline haemochromatosis diagnoses, with a cumulative incidence of 40.5% at age 80 years. There were excess incident liver disease (8.9% vs 6.8%; HR 1.62, 95% CI: 1.27 to 2.05, p=7.8×10⁻⁵), joint replacements and delirium, with similar results in the undiagnosed. p.C282Y/p.H63D and p.H63D+/+ men or women had no statistically significant excess fatigue or depression at baseline and no excess incident outcomes.

**Conclusions** Male and female p.C282Y homozygotes experienced greater excess morbidity than previously documented, including those undiagnosed with haemochromatosis in the community. As haemochromatosis diagnosis rates were low at baseline despite treatment being considered effective, trials of screening to identify people with p.C282Y homozygosity early appear justified.

## STRENGTHS AND LIMITATIONS OF THIS STUDY

⇒ We analysed large-scale data on community volunteers from the UK Biobank, one of the world's largest *HFE* genotyped cohorts.

⇒ We have analysed incident disease outcomes during an extended follow-up period of a mean of 13.3 years.

⇒ We have provided the first clinical outcome data to age 80 years in those with haemochromatosis genotypes, including those undiagnosed with haemochromatosis at baseline, expanding the life course evidence on *HFE* penetrance.

⇒ UK Biobank participants were somewhat healthier than the general population, but *HFE* allele frequencies were similar to previous UK studies.

⇒ Incident outcomes were from hospital inpatient and cancer registry follow-up, so it did not rely on potentially biased patient self-reporting, but community-diagnosed conditions may be underestimated.

## INTRODUCTION

*HFE* haemochromatosis is defined by iron overload[1] [2] due to gene variants, which dysregulate intestinal iron absorption. The p.C282Y+/+ (homozygous) group has markedly raised iron measures: for example, median transferrin saturations were over 80% and near 60% in men and women with p.C282Y+/+ in the Hemochromatosis and Iron Overload Screening (HEIRS) study but below 45% in compound heterozygotes (C282Y+/H36D+) and progressively lower across p.H63D+/+ (homozygote), p.C282Y± and p.H63D± carriers.[3] Women with each genotype have lower mean iron measures than men.[4]

Clinical presentation of haemochromatosis is usually with fatigue, joint pain or raised iron measures or from family screening or less commonly from direct-to-consumer genotyping. Symptoms usually present after the age of 40 years.[5] In severely affected patients

**Table 1** Baseline characteristics of male and female UK Biobank participants by selected p.C282Y/p.H63D genotypes

|  | No p.C282Y or p.H63D variants | p.H63D+/+ | p.C282Y+/p.H36D+ | p.C282Y+/+ |
|---|---|---|---|---|
| **Males** | | | | |
| Total participants | 122 841 | 4673 | 4959 | 1298 |
| Mean age, years (SD) | 56.99 (8.1) | 56.99 (8.1) | 56.97 (8.1) | 56.84 (8.2) |
| Haemochromatosis diagnosis, n (%) | 29 (0.02) | 8 (0.2) | 29 (0.6) | 157 (12.1) |
| Self-reported fatigue, n (%) | 12 094 (10.1) | 451 (9.9) | 523 (10.9) | 148 (11.8) |
| Self-reported fatigue (60+ years), n (%), | 4475 (8.2) | 173 (8.2) | 177 (8.0) | 68 (11.8) |
| Prevalent depression, n (%) | 5883 (4.8) | 220 (4.7) | 188 (3.8) | 71 (5.5) |
| **Females** | | | | |
| Total participants | 145 694 | 5580 | 5760 | 1604 |
| Mean age, years (SD) | 56.62 (7.9) | 56.58 (8.1) | 56.46 (7.9) | 56.92 (8.0) |
| Haemochromatosis diagnosis, n (%) | 8 (0.01) | <5 | 12 (0.2) | 54 (3.4) |
| Self-reported fatigue, n (%) | 19 110 (13.5) | 760 (14.1) | 785 (14.1) | 220 (14.4) |
| Self-reported fatigue (60+ years), n (%), | 6055 (10.0) | 253 (10.9) | 229 (9.9) | 77 (11.4) |
| Prevalent depression, n (%) | 10 734 (7.4) | 387 (6.9) | 432 (7.5) | 123 (7.7) |

A total of 451 270 male and female participants genetically similar to the 1000 Genomes Project European reference population with *HFE* genotypic data available in the UK Biobank. Numbers presented are mean (SD) for continuous variables and n (%) for categorical variables. Fatigue = tiredness in more than half the days in past 2 weeks. See online supplemental eTables 2 and 3 for all p.C282Y/p.H63D genotype groups.

with p.C282Y+/+ (>90% of typical cases[6]), liver iron deposition can lead to liver fibrosis, cirrhosis and cancer,[7] especially in the presence of other causes of liver disease. Arthropathy,[8] diabetes,[9] endocrine dysregulation,[10] heart arrhythmias and cardiomyopathies,[11] and pneumonia[9] have also been reported.

While clinical cohorts frequently have iron overload complications, disease penetrance in population genotyped groups is uncertain, especially at older ages and those not diagnosed with haemochromatosis.[3 12 13] Beutler *et al* found negligible haemochromatosis symptoms in 152 p.C282Y homozygotes from California health appraisal clinics (excluding diagnosed patients).[12] The HEIRS study reported excess liver disease in 299 men with p.C282Y+/+.[14] The Melbourne Collaborative Cohort Study[15] reported that 28.4% (95% CI, 18.8 to 40.2) of men with p.C282Y+/+ (n=95, mean age 65 years at follow-up) had 'documented iron overload-related disease', with 1.2% of women with p.C282Y+/+ affected. Similarly, although excess mortality occurred in clinical patients (especially with liver disease),[16 17] no excess mortality was reported in community-identified men with p.C282Y+/+ or other *HFE* genotype groups.[15 18] Using the UK Biobank, we previously examined data on European ancestry community participants with mean 7-year follow-up, finding that the 1294 male p.C282Y homozygotes had increased odds of liver disease and osteoarthritis compared with those without p.C282Y or p.H63D variants.[9] A UK Biobank study with an 8.9-year follow-up quantified excess hepatic malignancies in male p.C282Y homozygotes (HR 10.5; 95% CI: 6.6 to 16.7; p<0.001 vs no *HFE* variant) and excess all-cause mortality (n=88 deaths; HR 1.2; 95% CI: 1.0 to 1.5; p=0.046).[7]

For p.C282Y/p.H63D compound heterozygotes, the Melbourne Collaborative Cohort Study[15] found only one male (of 242 studied) with documented iron overload-related disease, although alcohol was also a factor. Similarly, a study of community-identified participants with p.H63D variants from the Busselton study (Australia) found none with clinically significant iron overloading.[19]

Given the accumulating evidence of significant clinical penetrance with p.C282Y homozygosity (only) and the reported effectiveness of treatment (predominantly venesection), there is renewed interest in screening for those at high risk.[1 20 21] *HFE* p.C282Y homozygosity is recommended for reporting to patients when found incidentally.[22]

The UK Biobank community cohort includes baseline questionnaires, plus hospital inpatient diagnoses during the mean 13.3-year follow-up. The cohort therefore provides a potential model for community-based genetic screening. Participant consent did not allow genotype feedback (see Methods), so outcomes reflect normal clinical care. Here, we aimed to estimate risks and cumulative outcomes to age 80 years by genotype and sex for relevant clinical outcomes, including (for the first time) analyses of those undiagnosed with haemochromatosis at baseline.

## METHODS
### Study population
The UK Biobank includes community volunteers aged 39–73 years at baseline assessments across England, Scotland and Wales, from 2006 to 2010. Participants were somewhat healthier than the general population,[23] but *HFE* allele frequencies were similar to previous UK

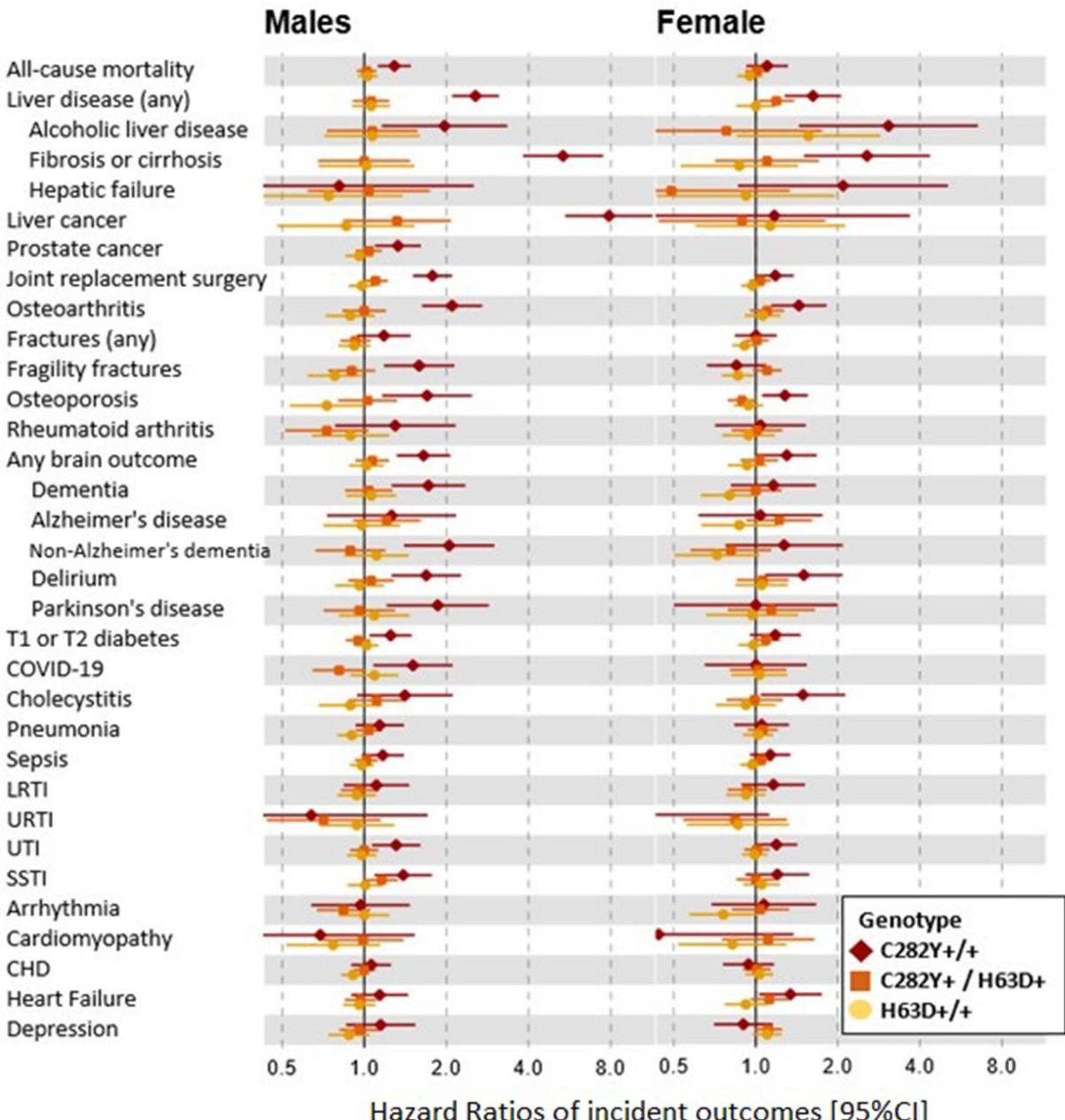

**Figure 1** HRs of incident disease outcomes (95% CI) in selected p.C282Y/p.H63D genotypes compared with those with no mutations. HRs compared with those with neither *HFE* mutation. Cox proportional hazard regression models adjusted for age, assessment centre and genetic principal components 1–10. Joint replacement surgery includes a diagnosis of hip, knee, ankle or shoulder replacement. Any brain outcome included a diagnosis of delirium, dementia or Parkinson's disease. See online supplemental eTables 4, 5, 9 and 10 for incident numbers and HRs for all p.C282Y/p.H63D genotype groups. CHD, coronary heart disease; LRTI, lower respiratory tract infection; URTI, upper respiratory tract infection; SSTI, skin and soft tissue infection; UTI, urinary tract infection.

studies.[7] Data cover 451 270 participants, genetically similar to the 1000 Genomes Project European reference population,[24] with *HFE* p.C282Y (rs1800562) and *HFE* p.H63D (rs1799945) genotypes from whole exome sequencing (whole exome sequence methods were by Regeneron[25]). Participants gave written informed consent and were informed of relevant health-related findings at baseline, but consent excluded individual notification of subsequent findings including genotypes. All research

was conducted in accordance with both the Declarations of Helsinki and Istanbul.

### Baseline variables and incident health outcomes

Baseline questionnaires covered doctor-diagnosed conditions including haemochromatosis. Symptom questions included: 'Over the past 2 weeks, how often have you felt tired or had little energy?', and responses were coded as 'fatigue' combining 'more than half the days' with

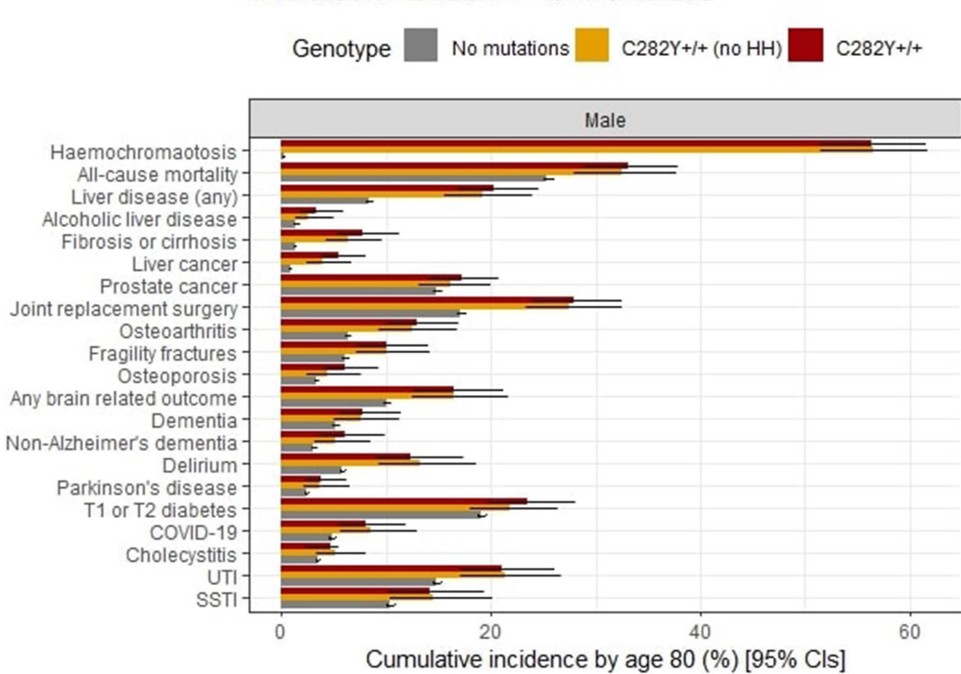

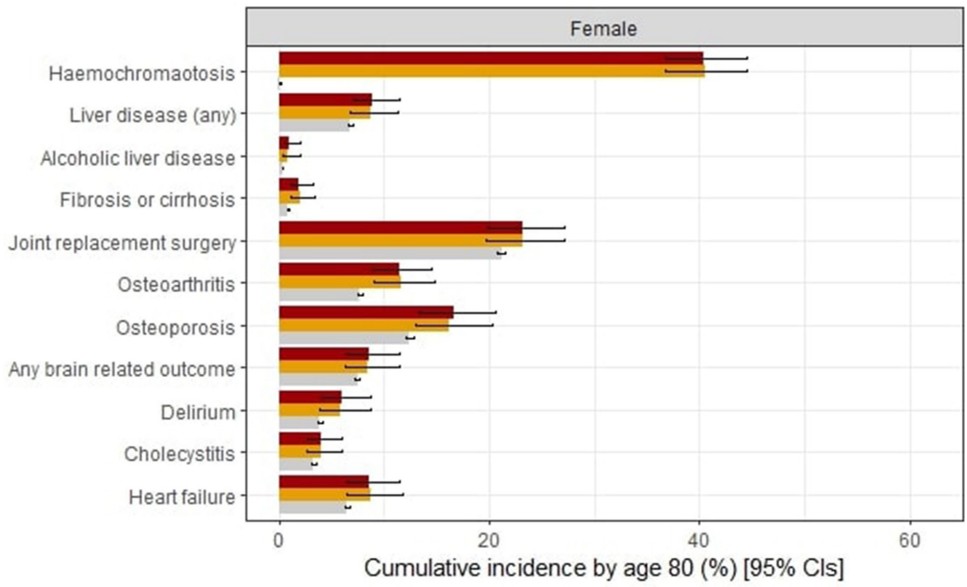

**Figure 2** Cumulative incidence of outcomes from ages 40 to 80 years by *HFE* genotypes (no p.C282Y or p.H63D variants vs p.C282Y homozygotes and p.C282Y homozygotes undiagnosed with haemochromatosis at baseline). Cumulative incidence (estimated % diagnosed by age 80 years; 95% CIs) for significant outcomes (p<0.05) from Cox proportional hazard regression models (figure 1; online supplemental eTables 5, 7, 10 and 12). Joint replacement surgery includes a diagnosis of hip, knee, ankle or shoulder replacement. Any brain outcome included a diagnosis of delirium, dementia or Parkinson's disease. SSTI, skin and soft tissue infection; UTI, urinary tract infection.

'nearly every day'. Studied diagnoses were from a priori knowledge (see online supplemental eTable 1 for ascertainment codes for 34 outcomes). England, Wales and Scotland hospital records were available from April 1996 to October 2022. Prevalent diagnoses were from baseline self-report plus hospital inpatient data from 1996 to

baseline. Incident diagnoses and surgical procedures were from hospital inpatient data (baseline to October 2022) plus cancer registries to December 2020 for England and Wales, and November 2021 for Scotland. National death records were available to November 2022. Disease ascertainment used International Classification of Diseases 10th revision codes. Surgical procedures were from Office of Population, Censuses and Surveys: Classification of Interventions and Procedures (OPCS-4). Having 'any joint replacement surgery' included hip, knee, ankle or shoulder replacement surgery. The 'any brain outcome' included delirium, dementia or Parkinson's disease.

## Statistical analysis

Cox proportional hazard regression estimated genotype associations with incident outcomes. Models were stratified by sex and adjusted for age, assessment centre and 10 genetic principal components (accounting for population substructure). Main outcome proportional hazard assumptions (tested using 'estat phtest') were met in models adjusted for 5-year age bands. Given the extensive prior evidence for risks in p.C282Y+/+ groups, multiple testing corrections are discussed for lower-risk genotypes, using Bonferroni correction of p<0.001 (0.05/34 disease outcomes). Kaplan-Meier survivor functions estimated the probabilities of cumulative incidence for associated outcomes from age 40 to 80 years within 5-year bands, by *HFE* genotype and by sex. We applied observed incidence rates in each age group to a notional cohort, estimating hypothetical cumulative incident case numbers from age 40 to 80 years. A sensitivity analysis repeated main analyses excluding participants with a haemochromatosis diagnosis at baseline. All analyses were performed in Stata 17.0.

## Patient and public involvement

Patients and participants were and are extensively involved in the UK Biobank study itself. We used anonymised data that were already collected, and therefore, no patients were involved in developing the research question or the outcomes tested. The UK Biobank notified participants of relevant health-related findings in the baseline assessment, but there is no individual notification of subsequent findings, including genotypes.

## RESULTS

UK Biobank baseline characteristics were previously reported[9 26] (table 1, online supplemental eTables 2 and 3): 451 270 participants genetically similar to the European 1000 Genomes reference population were followed for a mean of 13.3 years. There were 1298 men with p.C282Y+/+ (homozygotes), 4959 with p.C282Y/p.H63D compound heterozygote and 4673 with p.H63D+/+: for women 1604, 5760 and 5580, respectively. In the male p.C282Y+/+ group, 12.1% had haemochromatosis diagnoses at baseline, with 6/1000 diagnosed in p.C282Y/p.H63D and 2 per 1000 in the p.H63D+/+ group: the

respective rates in women were 3.4%, 2 per 1000 and 4 per 10 000.

Men with p.C282Y+/+ aged 60 years plus reported baseline excess fatigue (11.8% vs 8.2%; OR: 1.43, 95% CI: 1.11 to 1.85, p=0.01), but there was no statistically significant excess fatigue with other genotypes, except a marginal association in men with p.H63D± (OR: 1.06, 95% CI 1.00 to 1.12, p=0.05) which became non-significant after multiple testing correction. There were no differences in depression prevalence at baseline (table 1, online supplemental eTables 2 and 3).

### Men with p.C282Y homozygous genotypes: excess mortality and morbidity versus without *HFE* variants

Figure 1 shows the HRs for studied incident outcomes by sex and genotype, with cumulative incidence (with CIs) at age 80 years presented in figure 2 for those outcomes which had significant HRs. Men with p.C282Y+/+ had increased rates of mortality versus those without p.C282Y or p.H63D variants (HR 1.29, 95% CI: 1.12 to 1.48, p=$4.7{\times}10^{-4}$; figure 1; online supplemental eTables 4 and 5). A cumulative incidence of death was 33.1% (95% CI: 28.9 to 37.8) vs 25.4% (figures 2 and 3A). Excess mortality in those undiagnosed with haemochromatosis at UK Biobank baseline (online supplemental eTables 6 and 7) was similar (HR 1.22, 95% CI: 1.05 to 1.43, p=$1.0{\times}10^{-2}$) with a cumulative death rate of 32.5% (95% CI: 27.9 to 37.6) (online supplemental eFigure 1A).

Haemochromatosis diagnosis cumulative incidence at age 80 years in men with p.C282Y+/+ was 56.4% (figure 2, online supplemental eTable 8). Cumulative incidence of 'any liver disease' was 20.3% vs 8.3% without variants (HR 2.56, 95% CI: 2.10 to 3.12, p=$8.70{\times}10^{-21}$), and 7.7% developed liver fibrosis or cirrhosis versus 1.3%. There was a raised HR for alcoholic liver disease (figure 1) with a cumulative incidence 3.3% of men with p.C282Y+/+ versus 1.4%. Liver cancer cumulative incidence was 5.5% (95% CI: 3.8 to 8.0 vs 0.8% without variants). Men with p.C282Y+/+ also had raised a cumulative incidence of prostate cancer: 17.2% vs 14.8%. The cumulative incidence graphs show excess liver disease clearly apparent by age 55 years, but the excess mortality becomes significant at older ages (figure 3A,B). Notably, 81.5% of deaths, 66.3% of 'any liver diseases' and 71.8% of joint replacements in men with p.C282Y+/+ occurred after age 65 years.

Joint replacement cumulative incidence in men with p.C282Y+/+ was 27.9% vs 17.1% without variants (figure 3C): this remained after excluding those with fractures within 5 days before surgery (n=53; HR 1.74, 95% CI: 1.45 to 2.09; p=$1.6{\times}10^{-9}$). Men with p.C282Y+/+also had excess osteoarthritis, fragility fractures and osteoporosis.

Brain outcome (dementia, delirium or Parkinson's disease) cumulative incidence in men with p.C282Y+/+ was 16.4% vs 10.0% without variants (HR 1.65, 95% CI: 1.31 to 2.06, p=$1.70{\times}10^{-5}$) (figure 3D); for delirium, 12.4% vs 5.8% (HR 1.69, 95% CI: 1.26 to 2.27; p=$4.80{\times}10^{-4}$); for non-Alzheimer's dementia, 6.0% vs 3.0% (HR 2.05,

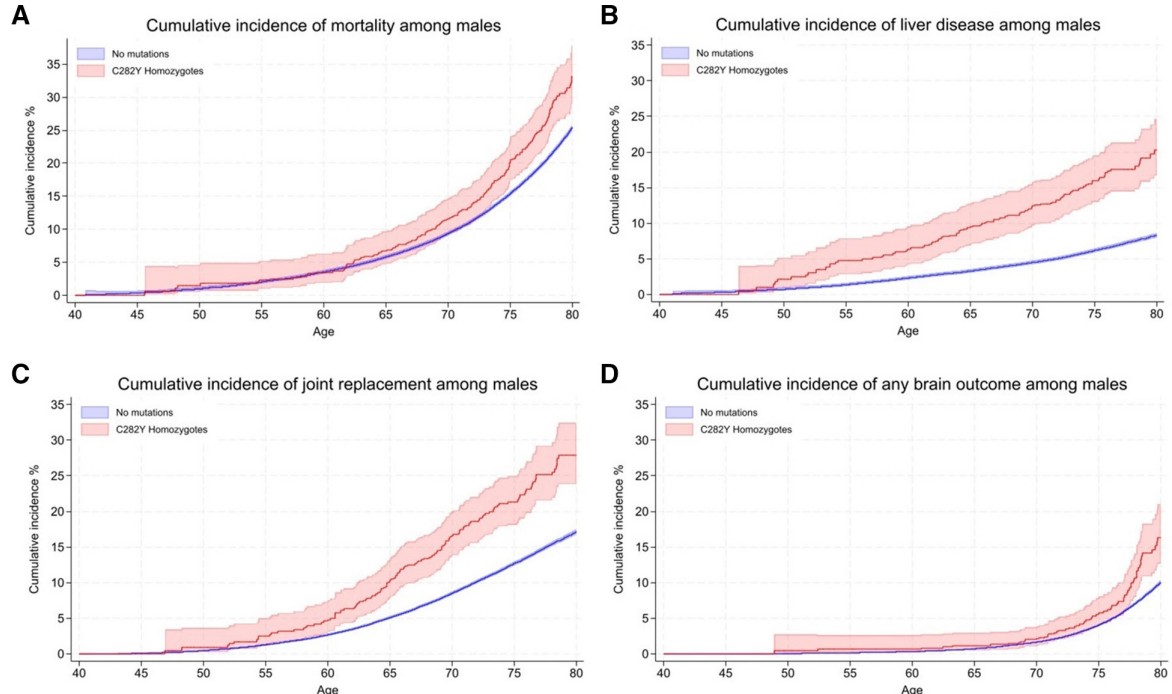

**Figure 3** Kaplan-Meier curves for the cumulative incidence of (A) mortality, (B) liver disease, (C) any joint replacement and (D) any brain outcome, in male *HFE* p.C282Y homozygotes compared with those with no mutations. Cumulative incidence (estimated % diagnosed by age 80 years; 95% CIs). Joint replacement surgery includes a diagnosis of hip, knee, ankle or shoulder replacement. Any brain outcome included a diagnosis of delirium, dementia or Parkinson's disease.

95% CI: 1.40 to 3.00; p=$2.40\times10^{-4}$); and for Parkinson's disease, 3.7% vs 2.4% (HR 1.86, 95% CI: 1.21 to 2.87 p=0.005).

Diabetes (type 1 or 2) cumulative incidence was 23.5% in men with p.C282Y+/+ (vs 19.1%). Men with p.C282Y+/+ had excess hospital-diagnosed COVID-19 (8% vs 4.8%, HR 1.51, 95% CI: 1.08 to 2.11; p=0.02, missing significance in earlier follow-up data[27]), urinary tract infections (UTIs) and skin and soft tissue infections (SSTIs). Associations were not significant for cardiac outcomes (arrhythmia, cardiomyopathy, coronary heart disease and heart failure) or depression.

Estimates of excess morbidity in the male p.C282Y+/+ group without haemochromatosis diagnoses in the community at UK Biobank baseline (n=1141) were mostly similar to those in the overall sample (figure 2, online supplemental eFigure 1A–D, online supplemental eTables 6 and 7). Notably, however, point HR estimates for liver fibrosis or cirrhosis and liver cancer risk appeared marginally lower than in the whole group but with wide CIs including the whole study estimates (eg, for liver fibrosis and cirrhosis, HR 4.52, 95% CI: 3.07 to 6.66, p=$2.10\times10^{-14}$ without a diagnosis, vs HR 5.36, 95% CI: 3.83 to 7.52; p=$2.00\times10^{-22}$ in the whole p.C282Y+/+ group). In addition, alcoholic liver disease and osteoporosis lost statistical significance, but cholecystitis became nominally statistically significant (HR 1.54, 95% CI: 1.02 to 2.32, p=0.04).

## Women with p.C282Y homozygous genotypes: excess morbidity versus without *HFE* variants

Diagnosed haemochromatosis cumulative incidence in women with p.C282Y+/+ was 40.5% by age 80 years. There was no excess mortality, but this group experienced excess 'any liver disease' (8.9% vs 6.75%; HR 1.62, 95% CI: 1.27 to 2.05; p=$7.80\times10^{-5}$) (figure 4A), liver fibrosis or cirrhosis (1.9% vs 0.8%; HR 2.56, 95% CI: 1.50 to 4.36; p=0.001), alcoholic liver disease (1.0% vs 0.3%; HR 3.07, 95% CI: 1.44 to 6.54; p=0.004) and cholecystitis, versus women without *HFE* variants (figures 1 and 2, online supplemental eTables 8–10).

Joint replacement surgery was more common in female p.C282Y homozygotes (23.2% vs 21.1%) (figure 4B), as were osteoarthritis and osteoporosis. A cumulative incidence of brain outcomes was raised (8.6% vs 7.4%; HR 1.30, 95% CI: 1.01 to 1.68; p=0.04), including delirium (5.9% vs 3.9%; HR 1.50, 95% CI: 1.08 to 2.08; p=0.02).

There was a nominally significant excess of heart failure (HR 1.34, 95% CI:1.03 to 1.75, p=0.03) but no associations with other studied cardiac outcomes. There were no associations with liver cancer, fragility fractures, diabetes, dementia, Parkinson's disease or infections (SSTIs and UTIs).

Estimates of excess morbidity in the female p.C282Y+/+ group without haemochromatosis diagnoses in the community at UK Biobank baseline (n=1550) were similar to those in the overall sample (figure 2, online supplemental eFigure 2A,B, online supplemental eTables 11 and 12).

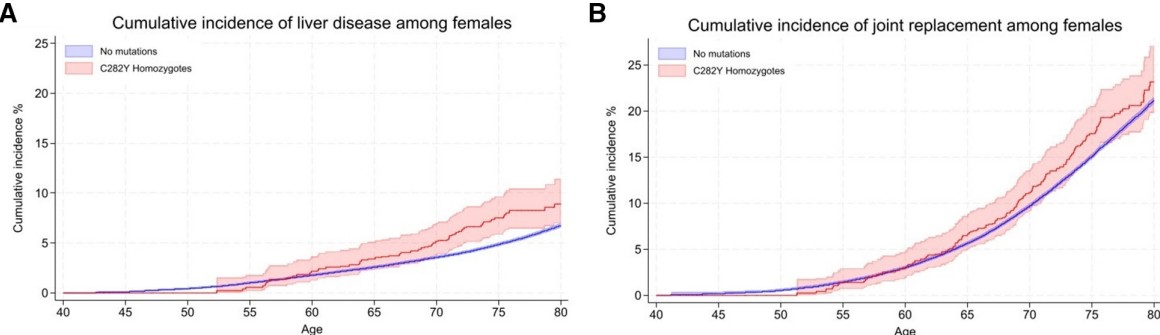

**Figure 4** Kaplan-Meier curves for the cumulative incidence of (A) liver disease and (B) any joint replacement, in female *HFE* p.C282Y homozygotes compared with those with no mutations. Cumulative incidence (estimated % diagnosed by age 80 years; 95% CIs). Joint replacement surgery includes a diagnosis of hip, knee, ankle or shoulder replacement.

## Men and women with p.C282Y/p.H63D compound heterozygous genotypes versus without *HFE* variants

Haemochromatosis diagnosis cumulative incidence in men with p.C282Y/p.H63D was 5.4% by age 80 years. However, there was no statistically significant increase in mortality, incidence of 'any liver disease' or liver cancers, joint replacements, or diabetes (figure 1). Men with p.C282Y/p.H63D were modestly more likely to develop SSTIs (HR 1.16, 95% CI: 1.01 to 1.33, p=0.03), but this lost statistical significance after multiple testing corrections.

Haemochromatosis cumulative incidence in women with p.C282Y/p.H63D was 2.7% by age 80 years, but there were similarly no excess mortality, musculoskeletal diagnoses, diabetes, or brain outcomes (figure 1). However, there was a modest association with 'any liver disease' (HR 1.19, 95% CI: 1.02 to 1.38, p=0.03), but associations with specific liver diagnoses were non-significant, including for liver fibrosis and cirrhosis, the most common form of liver disease with iron overload (figures 1 and 2, online supplemental eTables 4, 5 and 8–10).

## Men and women with p.H63D homozygous genotypes versus without *HFE* variants

Diagnosed haemochromatosis cumulative incidence in men with p.H63D/H63D was 1.9% by age 80 years but with no excess mortality, liver disease diagnoses, liver cancers, joint replacement surgery, diabetes or depression. There were protective associations between p.H63D homozygosity and fragility fractures (HR 0.78, 95% CI: 0.62 to 0.98 p=0.03) and osteoporosis (HR 0.73, 95% CI: 0.53 to 0.99, p=0.04) (online supplemental eTables 4, 5 and 8).

Diagnosed haemochromatosis cumulative incidence in women with p.H63D/H63D was 0.6% by age 80 years. There was no excess mortality, liver disease or liver cancers, joint replacement surgery, diabetes or depression. However, women also had a nominally significant (uncorrected p<0.05) protective association with fragility fractures (HR 0.87, 95% CI: 0.76 to 1.00 p=0.05), echoing the similar protective associations in men with p.H63D/H63D (online supplemental eTables 8–10).

## Associations for men and women with p.C282Y+/- and p.H63D+/- (heterozygote carrier)

There were a small number of nominally significant protective and risk associations in carrier groups (online supplemental eTables 4 and 5), in men with p.C282Y± including for excess delirium (HR 1.13, 95% CI: 1.04 to 1.24; p=0.006) and SSTIs (HR 1.14, 95% CI: 1.06 to 1.22; p=$1.80\times10^{-4}$) and men with p.H63D±, a protective association with fragility fractures (HR 0.92, 95% CI: 0.86 to 0.99; p=0.03). For women with p.C282Y±, there were no associations, but for women with p.H63D±, a nominally significant excess of liver cancers (HR 1.26, 95% CI: 1.01 to 1.58; p=0.04) was present, plus a protective association for osteoporosis (online supplemental eTables 9 and 10). Only the SSTI association was significant after multiple testing corrections.

## DISCUSSION

Clinical penetrance by *HFE* genotypes in community samples has been uncertain, especially for those undiagnosed with haemochromatosis, the potential beneficiaries of population screening. Our results show greater excess morbidity and mortality in p.C282Y+/+ groups than previously documented, much occurring at older ages. Men with p.C282Y+/+ had marked excess mortality (33.1% dead by age 80 years vs 25.4% without p.C282Y or p.H63D variants), plus substantial excess joint replacements, liver disease, brain outcomes and some infections. Estimates of excess morbidity in p.C282Y+/+ groups undiagnosed for haemochromatosis were similar (online supplemental eTables 7 and 12): perhaps most crucially, the p.C282Y homozygote male undiagnosed group had similar excess mortality, with an estimated cumulative death rate at age 80 years of 32.5% (95% CI: 27.9 to 37.6) vs 25.4% in those without *HFE* variants (online supplemental eFigure 1A, online supplemental eTable 7). Of women with p.C282Y+/+ (n=1604), diagnosed haemochromatosis cumulative incidence by 80 years old was 40.5%, with excess 'any liver disease', joint replacements and delirium present, suggesting more clinical penetrance than previously documented. In p.C282Y/p.H63D

and H63D homozygote groups and carriers, we found no excess fatigue or depression at baseline and no excess mortality or major outcomes during follow-up.

## Comparison of previous fatigue and depression studies

Baseline characteristics in the UK Biobank have been reported before,[26] except for fatigue and depression. Fatigue and depression in haemochromatosis are important to patients and their carers, but the causal role of iron overload in these symptoms is unclear. Men with p.C282Y+/+aged 60 years plus reported excess fatigue (11.8% vs 8.2% without *HFE* variants) at baseline but no excess depression at baseline or follow-up. Rates of both conditions were similar in the other studied genotype groups to those without *HFE* variants (except in men with p.H63D±, with very modest excess fatigue, non-significant with multiple statistical testing corrections). A blinded randomised trial of erythrocytapheresis[28] in p.C282Y+/+ groups with moderately raised ferritin levels (300 µg/L and 1000 µg/L) did find reduced fatigue scores with transferrin saturations falling from mean 63.5% to 45.4% in the treatment group, although the mean fatigue score was below diagnosis level, making clinical interpretation unclear. Also, recent observational studies have cast doubt on relationships between fatigue, quality-of-life measures and iron parameters in patients with haemochromatosis,[29 30] and fatigue might also occur with venesection. A UK Biobank analysis found tiredness genetically linked to factors including increased adiposity, blood lipids and inflammatory markers.[31]

## Comparison of previous mortality and morbidity studies

Rates of haemochromatosis diagnosis at baseline (mean age 57 years) were low even in the p.C282Y+/+ group (12.1% in men and 3.4% in women), but cumulative estimates at age 80 years indicated much higher diagnosis rates. These baseline rates are similar to cumulative rates from the US Electronic Medical Records and Genomics (eMERGE) network[32] for the p.C282Y+/+ group at age 60 years. The eMERGE study[32] figures also show the majority of haemochromatosis diagnoses occurring later in life, reaching nearly 50% in men and 25% in women after age 80 years.

Earlier reports suggested no excess mortality in community-identified p.C282Y+/+,[15 18] despite higher death rates in clinical cohorts with liver disease.[15 33] In the UK Biobank much larger sample, we found clearly increased all-cause mortality in men with p.C282Y+/+ only (194 p.C282Y+/+ deaths, HR 1.29, 95% CI: 1.12 to 1.48, p=$4.70\times10^{-4}$). A Swedish sample of 2273 p.C282Y homozygotes found similar increased mortality (HR 1.30, 95% CI: 1.12 to 1.50, vs men with no mutations), with no excess in female homozygotes (HR 0.98, 95% CI: 0.82 to 1.18).[34] These studies found no excess mortality in p.C282Y/p.H63D or p.H63D/p.H63D groups.

Estimates of disease penetrance in community genotyped groups differ widely,[3 12 13] with the highest estimate from the Melbourne Collaborative Cohort Study,[15] of 28.4% (95% CI: 18.8 to 40.2) of men with p.C282Y+/+

(n=95, mean age 65 years at follow-up) having 'documented iron overload-related disease' and 1.2% of female p.C282Y homozygotes being affected. In the UK Biobank, we found that diagnosed haemochromatosis cumulative incidence was 36.1% in men with p.C282Y+/+ and 21.2% of women with p.C282Y+/+ by age 65 years (online supplemental eTable 8), suggesting a larger burden of iron overload disease, especially in women with p.C282Y+/+. Liver cancers occurred in 5.5% (95% CI: 3.8 to 8.0) of men with p.C282Y+/+ by age 80 years, a slightly lower estimate than at age 75 years[7] (7.2%; 95% CI: 3.9 to 13.1) but within the earlier CIs. The estimate remains comparable with a meta-analysis[13] estimated lifetime incidence of severe liver disease (cirrhosis or hepatocellular carcinoma) of 9% (95% CI: 2.6 to 15.3) in untreated male *HFE* p.C282Y+/+ homozygotes.

Several studies have reported excess arthritis in male p.C282Y homozygote and, to a lesser extent, in female homozygotes.[9] Our results extend a previous 11.5-year follow-up,[35] now showing the full extent of later-life joint replacements in male p.C282Y homozygotes and providing a robust measure for severe joint damage in haemochromatosis.

Excess diabetes has previously been reported in male p.C282Y homozygous,[9] but other studies have suggested that *HFE* mutations do not have important pathophysiological consequences in patients with type 2 diabetes.[36] We found a modest diabetes excess in individuals with p.C282Y+/+ (figures 1 and 2). Overall, as a cumulative diabetes incidence was 23.5% in men with p.C282Y+/+vs 19.1% without *HFE* variants, non-iron factors appear to now contribute a majority of diabetes even in men with p.C282Y+/+. Although cardiac complications are often noted in haemochromatosis reviews,[11 37] there is little evidence linking community-identified p.C282Y homozygosity to these outcomes[38 39]: our finding of excess heart failure in women with p.C282Y homozygous genotype only (HR 1.34, 95% CI: 1.03 to 1.75, p=0.03) needs further study.

Our previous UK Biobank report on brain outcomes (mean 10.5 years of follow-up) found that men with p.C282Y+/+ had excess dementia diagnoses and iron deposition in key brain areas on MRI.[40] The current analysis also found higher incidence especially of non-Alzheimer's dementia in men with p.C282Y+/+ plus more delirium. Loughnan *et al*[41] reported that men with p.C282Y+/+ had more Parkinson's disease (OR 1.83; 95% CI: 1.19 to 2.80; p=0.006) in cross-sectional analyses of UK Biobank: our incident analyses provide a similar estimate for Parkinson's disease in men with p.C282Y+/+ (HR 1.86, 95% CI: 1.21 to 2.87; p=0.005).

In the HEIRS study,[3] men and women with community genotyped p.C282Y/p.H63D (compound heterozygote) had substantially lower transferrin saturation levels compared with men and women with p.C282Y+/+. In line with this, a cumulative incidence of haemochromatosis diagnosed by age 80 years was relatively low (men 5.4% and women 2.7%), and there was no excess mortality,

liver fibrosis and cirrhosis, joint replacements, diabetes or depression in these groups. We did find that women with p.C282Y/p.H63D had nominally significant small excess of any liver disease (HR 1.19, 95% CI: 1.02 to 1.38; p=0.03) although with no excess liver fibrosis and cirrhosis, no similar association in men with p.C282Y/p.H63D and with a non-significant multiple testing p value. Men with p.C282Y/p.H63D had a small excess of SSTIs (HR 1.16, 95% CI: 1.01 to 1.33, uncorrected p=0.03), again with non-significant multiple testing p value.

### Strengths and limitations

We analysed large-scale cohort data on community volunteers, providing the first outcome data to age 80 years. *HFE* allele frequencies were similar to other UK studies,[7] but the UK Biobank did recruit somewhat healthier participants than the general population,[23] so we have focused on incident outcomes during the mean of 13.3-year follow-up: we found no deviations from proportional hazard assumptions in Cox models. Outcomes for both the overall genotype groups and those undiagnosed with haemochromatosis at the UK Biobank baseline are presented. Previous studies have suggested that treated homozygous patients with p.C282Y do not have an increased mortality rate compared with the general population,[42] but numbers with haemochromatosis diagnoses at baseline were small in the current study and so outcomes in this group were not presented. Also, there were no data available on the diverse routes to diagnosis, so modelling treated outcomes would likely be confounded. Further work is needed on the effect of early diagnosis and treatment of haemochromatosis. Most outcomes were from hospital inpatient care, so community-diagnosed conditions may be underestimated, but findings are similar to analyses including primary care records (available to 2017 only)[7] and baseline data collected at interview.[9] Information on the criteria used to establish a diagnosis of fibrosis/cirrhosis in the hospital follow-up data (eg, non-invasive biomarker panels or liver biopsy) were not available.

### Implications for early diagnosis and screening

We found greater excess morbidity than previously reported in both men and women p.C282Y+/+, especially at older ages, and in those undiagnosed with haemochromatosis. Despite treatment (predominantly venesection) being considered effective for preventing liver disease progression,[1 43 44] rates of haemochromatosis diagnosis were low at baseline: for example, of 31 men with p.C282Y+/+, with incident liver cancer, 17 were undiagnosed with haemochromatosis at baseline. Genotyping earlier in life could be a powerful preventive tool to identify individuals with p.C282Y+/+ who are clearly at major risk of related excess disease. A recent cross-sectional genotyping study within a US health provider found that 72% (144/201) of homozygous patients with p.C282Y (mean age 62 years) had not been diagnosed with haemochromatosis, 36% of whom had iron overload.[21] As cumulative diagnosed haemochromatosis in the UK Biobank cohort was estimated at 56.4% in men with p.C282Y+/+ and 40.5% in women by age 80 years with, p.C282Y+/+ genotyping with follow-up iron studies would likely have a high yield. However, the prevention potential following up p.C282Y/p.H63D and H63D+/+ groups will likely be very low, as there was no significant excess fatigue or depression at baseline and no excess mortality, incident liver fibrosis or cirrhosis, joint replacements, or depression.

### CONCLUSION

Men and women with p.C282Y homozygous genotypes in this community cohort experienced greater excess morbidity than previously documented. Much of this excess liver, musculoskeletal, diabetes, brain and morbidity occurred after the age of 60 years. In those not diagnosed with haemochromatosis at study baseline, risks were broadly similar to those in the overall group, notably including excess mortality in men with p.C282Y+/+. Trials of targeted or community genotyping to diagnose haemochromatosis earlier appear justified, especially to identify people with the p.C282Y homozygote variants. The potential for iron-related preventive treatment in the p.C282Y/p.H63D, p.H63D+/+ and other *HFE* genotype groups appears very limited, as these genotype groups were associated with no statistically significant excess morbidity or mortality.

**Acknowledgements** This research was conducted using the UK Biobank resource, under application 14631. We thank the UK Biobank participants and coordinators. This work used data provided by patients and collected by the NHS as part of their care and support. Reused with the permission of the NHS England and UK Biobank. All rights reserved. This research also used data assets made available by the National Safe Haven as part of the Data and Connectivity National Core Study, led by the Health Data Research UK in partnership with the Office for National Statistics and funded by the UK Research and Innovation. This study was supported by the National Institute for Health and Care Research Exeter Biomedical Research Centre. The views expressed are those of the authors and not necessarily those of the National Institute for Health and Care Research (NIHR) or the Department of Health and Social Care.

**Contributors** MRL performed the analysis, interpreted results, created the figures and drafted the manuscript. JLA contributed to the design and analysis of the study, data interpretation and drafting of the manuscript, and acted as guarantor. LP contributed to the design of the study, data interpretation and creation of figures and contributed to the manuscript. JS provided expert clinical interpretation of the data and contributed to the manuscript. DM oversaw the design of the study, data analysis, interpretation of results, and led the writing of the manuscript. All authors approved the final version of the manuscript.

**Funding** The University of Exeter supports MRL, LCP and DM; JLA has a NIHR Advanced Fellowship (NIHR301844).

**Competing interests** None declared.

**Patient and public involvement** Patients and/or the public were not involved in the design, or conduct, or reporting, or dissemination plans of this research.

**Patient consent for publication** Not applicable.

**Ethics approval** This study involves human participants and was approved by the North West Multi-centre Research Ethics Committee (Research Ethics Committee reference 11/NW/0382), which approved the UK Biobank. Participants gave informed consent to participate in the study before taking part.

**Provenance and peer review** Not commissioned; externally peer reviewed.

**ORCID iDs**
Janice L Atkins http://orcid.org/0000-0003-4919-9068
Luke C Pilling http://orcid.org/0000-0002-3332-8454

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
