## [Reviewer comments · BMJ Open]

ARTICLE DETAILS

TITLE (PROVISIONAL)	HFE genotypes, haemochromatosis diagnosis and clinical outcomes to age 80: a prospective cohort study in UK Biobank
AUTHORS	Lucas, Mitchell; Atkins, Janice; Pilling, Luke; Shearman, Jeremy; Melzer, David

VERSION 1 – REVIEW

REVIEWER	Olynyk, John Fiona Stanley Hospital, Gastroenterology and Hepatology
REVIEW RETURNED	27-Nov-2023

GENERAL COMMENTS	Overall a timely, well conducted review providing important clinical information. I have several comments that should be addressed in the revised version. Comments 1. Type-1 haemochromatosis is an old terminology that has been superseded by the term “HFE haemochromatosis”. Please adjust.2. Direct to consumer genotyping is not a common modality leading to clinical presentation in practice in most constituencies. I would remove it or qualify the statement to indicate this.3. Are there specific details available on the criteria used to establish a diagnosis of cirrhosis or fibrosis e.g. noninvasive biomarker panels or liver biopsy?4. Is there an ability to do a subgroup study on diagnosed and treated C282Y +/- individuals versus those who remained untreated to determine if the prevalence of delirium or dementia was lower in the treated group?5. The work of Davis et al Diabetes Care 2008 Sep;31(9):1795-80 should be referred to in relation to the prevalence of HFE mutations in subjects with type 2 diabetes mellitus.
--

REVIEWER	Bardou-Jacquet, Edouard Rennes 1 University
REVIEW RETURNED	04-Dec-2023

GENERAL COMMENTS	In this study, the authors aimed to assess the long-term clinical penetrance of the HFE genotype in the UK Biobank. They had previously published results regarding clinical penetrance and malignancy incidence over a slightly shorter time frame. In this report, they provide an in-depth analysis of penetrance and mortality up to the age of eighty, focusing specifically on morbidity in patients with and without a diagnosis of hemochromatosis at baseline. The results revealed a higher-than-anticipated clinical penetrance, with increased mortality and frequent bone-related outcomes observed in C282Y homozygous patients. Additionally, the study found that nearly half of the C282Y homozygous patients remained
--

	undiagnosed at the age of eighty. Furthermore, the results confirmed a higher prevalence of brain-related outcomes in C282Y homozygous patients, while also affirming the lack of clinical significance of other HFE genotypes. Despite several results having been previously published based on shorter follow-ups, this study contributes novel and significant findings. The large study population, rigorous methodology, and well-crafted manuscript enhance the credibility of the research. One aspect that warrants further discussion is the apparent lack of benefit from early diagnosis. Since venesection treatment is presumed to be beneficial, one would expect lower morbidity in diagnosed patients compared to those undiagnosed. Some studies even suggest improved survival compared to the general population (Bardou-Jacquet, J Hepatol, 2015). If there is no discernible benefit from early diagnosis, the relevance of community screening may be diminished. Minor Point: Page 6, Line 56: "In the male p.C282Y+/+ group, 12.1% had hemochromatosis diagnoses at..." Is there a missing word: diagnosis?
--	---

VERSION 1 – AUTHOR RESPONSE

Reviewer: 1 Prof. John Olynyk, Fiona Stanley Hospital Overall a timely, well conducted review providing important clinical information. I have several comments that should be addressed in the revised version. Comments	We thank the reviewer for their positive comments on our manuscript and have responded in detail below.	N/A
1. Type-1 haemochromatosis is an old terminology that has been superseded by the term "HFE haemochromatosis". Please adjust.	The term has been replaced throughout the manuscript.	Abstract, Line 1 Introduction, Line 1
2. Direct to consumer genotyping is not a common modality leading to clinical presentation in practice in most constituencies. I would remove it or qualify the statement to indicate this.	We have updated this sentence accordingly to qualify that direct-to-consumer genotyping is not a common modality leading to clinical presentation in practice: "Clinical presentation of haemochromatosis is usually with fatigue, joint pain, raised iron measures or from family screening, or less commonly from direct-to-consumer genotyping."	Introduction, Paragraph 2
3. Are there specific details available on the criteria used to	Incident diagnosis of cirrhosis/fibrosis were from	

establish a diagnosis of cirrhosis or fibrosis e.g. noninvasive biomarker panels or liver biopsy?	hospital inpatient data, in follow-up from baseline to October 2022 (ICD-10 code K74), as outlined in the Methods section. Unfortunately, information on the criteria used to establish a diagnosis of cirrhosis/fibrosis in the hospital data (e.g., non-invasive biomarker panels or liver biopsy) were not available. We have added a comment on this to the Discussion: “Information on the criteria used to establish a diagnosis of fibrosis/cirrhosis in the hospital follow-up data (e.g. non-invasive biomarker panels or liver biopsy) were not available.”	Discussion, Strengths and Limitations
4. Is there an ability to do a subgroup study on diagnosed and treated C282Y +/- individuals versus those who remained untreated to determine if the prevalence of delirium or dementia was lower in the treated group?	Unfortunately, numbers with diagnosed haemochromatosis at baseline were small (n=157 in male C282Y homozygotes and n=54 in female C282Y homozygotes). Also, information on routes to diagnosis and whether individuals were receiving treatment was unavailable. So, performing the suggested subgroup analysis would likely be confounded. We have updated the section in the Discussion to clarify: “Outcomes for both the overall genotype groups and those undiagnosed with haemochromatosis at UK Biobank baseline are presented. Previous studies have suggested that treated p.C282Y homozygous patients do not have an increased mortality rate compared to the general population (42), but numbers with haemochromatosis diagnoses at baseline were small in the current study and so outcomes in this group were not presented. Also, there was no data available on the diverse routes to diagnosis, so modelling treated outcomes would likely be confounded. Further work is needed on the effect so early diagnosis and treatment.”	Discussion, Strengths and Limitations
5. The work of Davis et al Diabetes Care 2008 Sep;31(9):1795-80 should be referred to in relation to the prevalence of HFE mutations in subjects with type 2 diabetes mellitus.	We thank the reviewer for this suggestion and have now referenced this study in the manuscript: “Excess diabetes has previously been reported in p.C282Y homozygous men (9), but other studies have suggested that HFE mutations do not have important pathophysiological consequences in patients with type 2 diabetes(36).”	Discussion, page 11 paragraph 5 Reference 36

	36. Davis TME, Beilby J, Davis WA, Olynyk JK, Jeffrey GP, Rossi E, et al. Prevalence, Characteristics, and Prognostic Significance of HFE Gene Mutations in Type 2 Diabetes. Diabetes Care. 2008 Sep 1;31(9):1795–801.	
Reviewer: 2 Dr. Edouard Bardou-Jacquet, Rennes 1 University In this study, the authors aimed to assess the long-term clinical penetrance of the HFE genotype in the UK Biobank. They had previously published results regarding clinical penetrance and malignancy incidence over a slightly shorter time frame. In this report, they provide an in-depth analysis of penetrance and mortality up to the age of eighty, focusing specifically on morbidity in patients with and without a diagnosis of hemochromatosis at baseline. The results revealed a higher-than-anticipated clinical penetrance, with increased mortality and frequent bone-related outcomes observed in C282Y homozygous patients. Additionally, the study found that nearly half of the C282Y homozygous patients remained undiagnosed at the age of eighty. Furthermore, the results confirmed a higher prevalence of brain-related outcomes in C282Y homozygous patients, while also affirming the lack of clinical significance of other HFE genotypes. Despite several results having been previously published based on shorter follow-ups, this study contributes novel and significant findings. The large study population, rigorous methodology, and well-crafted manuscript enhance the credibility of the research. One aspect that warrants further discussion is the apparent lack of benefit from early diagnosis. Since	We thank the reviewer for their positive comments on our manuscript. We did not present results in diagnosed patients as numbers with haemochromatosis diagnoses at baseline were small, and there was no data available on the diverse routes to diagnosis, or on whether individuals were undergoing treatment, so modelling treated outcomes would likely be confounded. Further work is needed in this area. We have added a note on this to the Discussion section, and included the suggested reference: “Outcomes for both the overall genotype groups and those undiagnosed with haemochromatosis at UK Biobank baseline are presented. Previous studies have suggested that treated p.C282Y homozygous patients do not have an increased mortality rate compared to the general population (42), but numbers with haemochromatosis diagnoses at baseline were small in the current study and so outcomes in this group were not presented. Also, there was no data available on the diverse routes to diagnosis, so modelling treated outcomes would likely be confounded. Further work is needed on the effect so early diagnosis and treatment.” 42. Bardou-Jacquet E, Morcet J, Manet G, Lainé F, Perrin M, Jouanolle AM, et al. Decreased cardiovascular and extrahepatic cancer-related mortality in treated patients with mild HFE hemochromatosis. J Hepatol. 2015 Mar;62(3):682–9.	Discussion, Strengths and Limitations Reference 42

venesection treatment is presumed to be beneficial, one would expect lower morbidity in diagnosed patients compared to those undiagnosed. Some studies even suggest improved survival compared to the general population (Bardou-Jacquet, J Hepatol, 2015). If there is no discernible benefit from early diagnosis, the relevance of community screening may be diminished.		
Minor Point: Page 6, Line 56: "In the male p.C282Y+/+ group, 12.1% had hemochromatosis diagnoses at..." Is there a missing word: diagnosis?	We have now updated this sentence to include the missing word: "In the male p.C282Y+/+ group, 12.1% had haemochromatosis diagnoses at baseline"	Results, Page 6, Paragraph 1

VERSION 2 – REVIEW

REVIEWER	Olynyk, John Fiona Stanley Hospital, Gastroenterology and Hepatology
REVIEW RETURNED	10-Jan-2024

GENERAL COMMENTS	All my concerns have been well addressed and I congratulate the authors on the study
--

REVIEWER	Bardou-Jacquet, Edouard Rennes 1 University
REVIEW RETURNED	12-Jan-2024

GENERAL COMMENTS	The authors adequately answered the raised question.
--